# Low-cost anti-mycobacterial drug discovery using engineered *E. coli*

Nadine Bongaerts [1,2], Zainab Edoo[3], Ayan A. Abukar [1,2], Xiaohu Song[1,2], Sebastián Sosa-Carrillo[1,4], Sarah Haggenmueller [1,2], Juline Savigny[1,2], Sophie Gontier [1,2], Ariel B. Lindner [1,2✉] & Edwin H. Wintermute[1,2✉]

Whole-cell screening for *Mycobacterium tuberculosis* (*Mtb*) inhibitors is complicated by the pathogen's slow growth and biocontainment requirements. Here we present a synthetic biology framework for assaying *Mtb* drug targets in engineered *E. coli*. We construct Target Essential Surrogate *E. coli* (TESEC) in which an essential metabolic enzyme is deleted and replaced with an *Mtb*-derived functional analog, linking bacterial growth to the activity of the target enzyme. High throughput screening of a TESEC model for *Mtb* alanine racemase (Alr) revealed benazepril as a targeted inhibitor, a result validated in whole-cell *Mtb*. In vitro biochemical assays indicated a noncompetitive mechanism unlike that of clinical Alr inhibitors. We establish the scalability of TESEC for drug discovery by characterizing TESEC strains for four additional targets.

[1] Université Paris Cité, Inserm, System Engineering and Evolution Dynamics, Paris, France. [2] CRI, Paris, France. [3] Sorbonne Université, Université Paris Cité, Inserm, Centre de Recherche des Cordeliers (CRC), Paris, France. [4] Present address: Institut Pasteur, Inria de Paris, Université Paris Cité, InBio, Paris, France. ✉email: ariel.lindner@inserm.fr; jake.wintermute@cri-paris.org

Chemical genetics in the synthetic biology era offers new tools for long-standing challenges in antimicrobial drug discovery[1]. With genetic modifications to targeted functional pathways, microbial strains can be sensitized to drugs of a particular mechanism[2]. High-throughput screening of these strains may then reveal new drugs that act specifically on the modified pathway. Recent successful applications of chemical genetics have produced drug scaffolds and helped to identify drug mechanisms in a variety of microbial pathogens[3–9].

The chemical-genetic strategy seeks to combine the advantages of two classical approaches to drug discovery: whole-cell screens that test directly on live pathogenic bacteria[10] and target-based screens that assay against purified cellular components[11]. Like whole-cell screening, it prioritizes molecules that can pass the membrane barrier and function in the rich intracellular context. Like target-based screening, it allows researchers to focus on specific biological functions of therapeutic interest and simplifies the process of determining a hit's mechanism of action.

Unique technical challenges arise when using genetically engineered microbes for antimicrobial discovery. The relationship between a target's activity and the host microbe's sensitivity to a targeted inhibitor is quantitative and pathway-dependent[12]. Empirical fine-tuning is required to maximize assay sensitivity while minimizing stress associated with gene over- or under-expression[13]. Precise expression control is difficult to achieve in many pathogens, for which genetic tools are often limited[14]. Finally, the conventional challenges of pathogen microbiology still apply. Long incubation times, noisy measurements, and expensive safety practices limit the scale at which these tools can be deployed[15].

Here we perform chemical-genetic screens for targeted inhibitors of pathogen-derived enzymes in an *E. coli* host. We develop TESEC strains in which an essential *E. coli* enzyme is deleted and replaced with a functionally equivalent target enzyme of heterologous origin. The target's induction is quantitatively controlled to determine the highest and lowest expression levels compatible with robust growth. Target-specific inhibitors are expected to be more effective against TESEC strains under low induction, forming the basis for a differential screen.

As a proof of concept, we developed a TESEC strain for the enzyme Alr derived from the human pathogen *Mtb*. A screen of 1280 approved drugs revealed benazepril as an expression-dependent inhibitor. Whole-cell antibacterial activity of benazepril was confirmed in both *Mycobacterium smegmatis* and *Mtb*. Biochemical assays revealed a non-competitive mechanism of inhibition unlike that of known Alr inhibitor D-cycloserine (DCS).

Benazepril is an off-patent ACE inhibitor widely used to control hypertension[16]. A retrospective study of the Taiwan national health insurance research database associated drugs of this class with a reduced risk of developing active tuberculosis[17]. Our results suggest that this outcome may be the result of an antimycobacterial activity of benazepril.

The TESEC system is designed for maximum versatility and re-use with Golden Gate assembly[18]. In principle, over 100 known conditionally essential *E. coli* metabolic genes could be complemented with pathogen-derived analogs and screened with TESEC[19]. We characterize here four additional TESEC strains for diverse metabolic targets in *Mtb*. Low-cost and biosafe screening technologies may allow wider participation in antibiotic discovery efforts and unlock new economic models to incentivize work in this long-neglected domain[20,21].

## Results

### Design and characterization of a TESEC strain for *Mtb* Alr.
Our interest in *Mtb* was motivated by the scope of the public health challenge it presents and by a decades-long gap in the emergence of new treatments[22]. Alr is a well-known drug target required for cell wall biosynthesis in both *Mtb*[23] and *E. coli*[24]. The enzyme catalyzes the reversible conversion of L-alanine to D-alanine, supplying an essential building block for peptidoglycan cross-linking[25]. DCS is a widely used Alr inhibitor that served as a positive control for our assays. Although an effective antibacterial, DCS is reserved as a second-line *Mtb* treatment due to serious neurological side effects[26]. DCS shows no cross-resistance with first-line tuberculosis drugs, highlighting the value of targeting Alr by other means[27].

To produce the TESEC Host Alr- strain, endogenous *E. coli* alanine racemase activity was eliminated by deletion of the genes *dadX* and *alr*, conveying a growth requirement for supplemental D-alanine. To maximize host sensitivity to screened compounds, we further deleted the *tolC* efflux system and the *entC* enterobactin synthase, to rescue a growth defect associated with *tolC* deletion (Supplementary Fig. 1).

Heterologous expression of *Mtb*-derived Alr was achieved with an arabinose-inducible, feedback-controlled circuit adapted from Daniel et al.[28]. Briefly, the system consists of a low-copy plasmid expressing AraC and a high-copy plasmid expressing *Mtb* Alr (Fig. 1a). Flow cytometry using GFP-tagged *Mtb* Alr confirmed that the circuit allowed uniform and unimodal protein expression over a wide dynamic range (Fig. 1b). Growth inhibition by DCS could be quantitatively controlled with half-maximal inhibitory concentrations ($IC_{50}$) ranging from 2 μM to 1 mM (Fig. 1c).

We chose low and high Alr induction levels by comparing resistance levels to DCS at 100 μM, a common working concentration for high-throughput drug screening (Fig. 1d). Low Alr induction (0.1 μM arabinose) was selected as the highest level that still supported complete growth repression under DCS treatment. High Alr induction (10 mM arabinose) was the highest level not associated with overexpression toxicity. The $IC_{50}$ of DCS varied by 50 fold between high- and low-induction conditions, suggesting a strong differential sensitivity to Alr.

### Differential screening for *Mtb* Alr inhibitors.
The TESEC *Mtb* Alr expression strain was screened against the Prestwick Chemical Library, 1280 approved small molecules suitable for drug repurposing (Fig. 2). The screen was conducted in triplicate for both low and high Alr induction levels, using defined media with compounds at 0.1 mM and 1% DMSO. Strain growth was measured by optical density (OD) at 600 nm after 10 h. High- and low-induction treatments with the positive control compound DCS produced a differential Z-factor of 0.87, indicating a robust assay.

Hits were determined to be drugs significantly inhibiting the growth of low Alr strains (OD < 0.1) but not high Alr (OD > 0.2) (Fig. 2a). The statistical significance of hits was assessed by strictly standardized mean difference (SSMD) to DMSO-only controls (Fig. 2b).

Ten compounds met differential growth inhibition and significance criteria (Fig. 2c). Of these, 7 belong to the β-lactam class of known antibiotics. Because both β-lactams and Alr act in the pathway of peptidoglycan synthesis[29], under-expression of Alr may fragilize the cell wall and therefore sensitize cells to β-lactam treatment, an interaction observed in other microbes[30]. The hit compounds benazepril and amlexanox had no previously reported antibacterial properties and so were retained for closer analysis.

We next repeated the screen for a range of Alr induction levels (Fig. 2d). The result is a detailed chemical-genetic sensitivity profile for each of the 1280 screened compounds. Globally, increased Alr expression was associated with reduced growth. DCS and benazepril, but not amlexanox, produced distinct

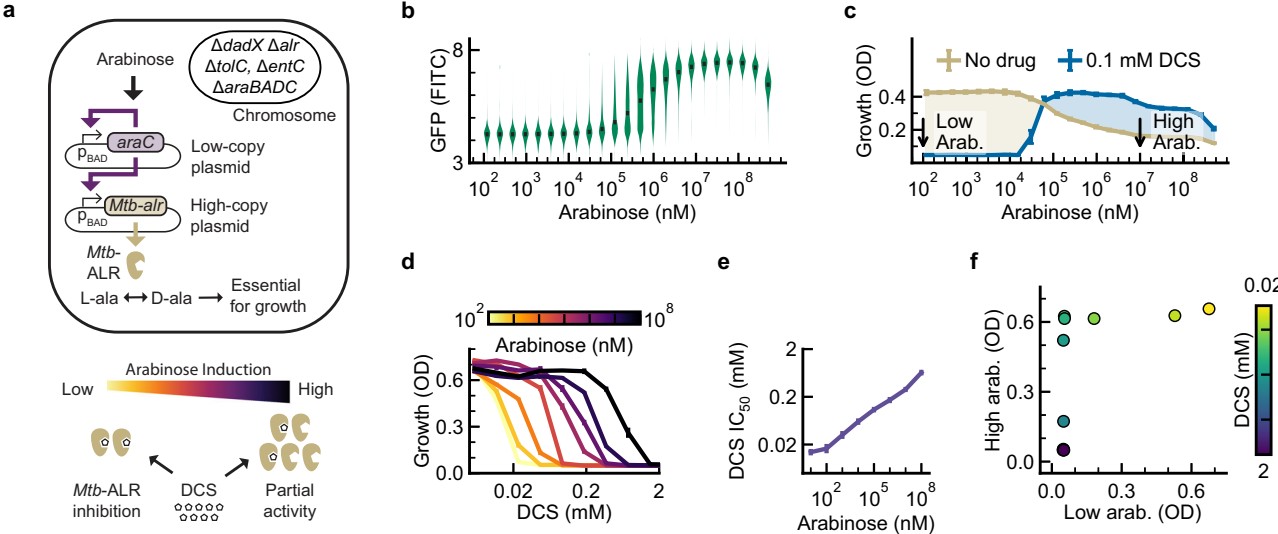

**Fig. 1 A TESEC strain for *Mtb* Alr shows differential sensitivity to targeted inhibitors. a** The TESEC host carries deletions of native Alr enzymes, the TolC efflux system, and genes participating in the native arabinose response. Two plasmids carry an arabinose-inducible, feedback-controlled expression circuit for *Mtb* Alr that produces a smoothed, log-linear induction response[28]. **b** A violin plot of flow cytometry data of *Mtb* Alr tagged with GFP. Expression was unimodal and log-linear for arabinose concentrations from $10^5$ to $10^7$ nM. **c** The effect of arabinose on TESEC growth in the presence or absence of 100 µM DCS. Shaded areas indicate the differential growth response to DCS. The indicated high and low arabinose concentrations were selected for use for screening. Data are presented as the mean and CI95 of 8 biological replicates. **d** TESEC *Mtb* Alr dose-response to DCS for a range of arabinose concentrations. Error bars are CI95 of 4 biological replicates. **e** The half-maximal effective dose of DCS varied by more than 50-fold as the arabinose induction level was varied. Error bars are CI95 of best-fit parameter estimates. **f** A mock differential screen of high- and low- induction TESEC for a range of DCS concentrations. Simulated hits appear as off-diagonal points where growth inhibition is observed for only the low-induction strain. Source data are provided as a Source Data file.

chemical-genetic profiles with improved growth under high Alr induction (Fig. 2e). Therefore, benazepril was retained for further validation.

**Target abundance affects chemical-genetic sensitivity profiles.** TESEC screening seeks to detect a phenotypic difference in drug sensitivity caused by changing the expression level of a drug's protein target, a strategy common to many chemical-genetic assays[3–8]. However, previous work has shown that perturbations in protein expression can also induce systemic stress responses caused by protein misfolding, resource depletion, or metabolic imbalance[13,31]. This could result in the identification of non-specific hit compounds with differential activity against stressed cells but not against the desired target. We therefore sought to better characterize the effect of Alr expression levels on the reliability of chemical-genetic drug discovery.

We compared TESEC screening for a range of Alr expression levels to an Alr+ wild-type control strain with intact native Alr activity and carrying expression plasmids for GFP only (Fig. 3). At low induction levels, the TESEC strain grew nearly as well as the wild-type control and responded similarly to drug treatments (Fig. 3a). Increasing Alr induction resulted in decreased overall growth and decreased correlation between TESEC and wild-type chemical-genetic profiles.

The Prestwick library includes 183 known antibiotics, many of which were effective against *E. coli* under our screening conditions. Higher Alr induction levels resulted in less statistical power for distinguishing growing from non-growing cells, therefore making antibiotic activities harder to detect (Fig. 3b). Interestingly, the antibiotic activity of DCS itself was not detectable in the wild-type strain because the effective concentration of the drug exceeds the 0.1 mM level commonly used in high-throughput screening[32]. Only low-induction Alr TESEC strains could correctly re-discover this known antibiotic.

Overexpression of Alr resulted in reduced screening sensitivity to known antibiotics (Fig. 3c). The whole-cell assay conducted on the Alr+ wild-type control strain identified 314 growth-inhibiting compounds of which 94 were described antibiotics, a true positive rate of 30%. Low-induction TESEC strains performed similarly while high-induction strains were nonspecifically inhibited by many compounds, resulting in false positive rates above 70%.

**Characterization of hits in TESEC.** The TESEC system provides an in-line platform to characterize and validate hit compounds before advancing them to biochemical or live-pathogen assays (Fig. 4). We produced two-dimensional chemical-genetic profiles of TESEC growth under a range of drug treatments and Alr expression levels (Fig. 4a–c). As expected, the inhibitory dose of DCS varied quantitatively with Alr level, producing a characteristic diagonal line in the growth heatmap. Benazepril produced a similar diagonal profile, consistent with a target-specific effect of benazepril, while amlexanox did not.

At higher resolution, benazepril and DCS produced qualitatively different chemical-genetic profiles. Treatment with DCS shifted the growth curve horizontally, altering primarily the Alr induction level required to produce resistance. Benazepril treatment shifted the growth curves vertically, reflecting similarly decreased growth across a range of Alr expression levels. Unlike DCS, the effect of benazepril treatment was only evident at low Alr induction. These results suggested different mechanisms of action for DCS and benazepril.

Supplementation of the growth medium with 5 mM D-alanine, the metabolic product of Alr, was able to rescue TESEC strain growth under treatment with both 0.25 mM DCS and 1 mM benazepril (Fig. 4d). This is consistent with a target-specific activity for the drugs.

Restoring activity of the efflux pump TolC eliminated benazepril sensitivity, indicating that the drug can be effluxed

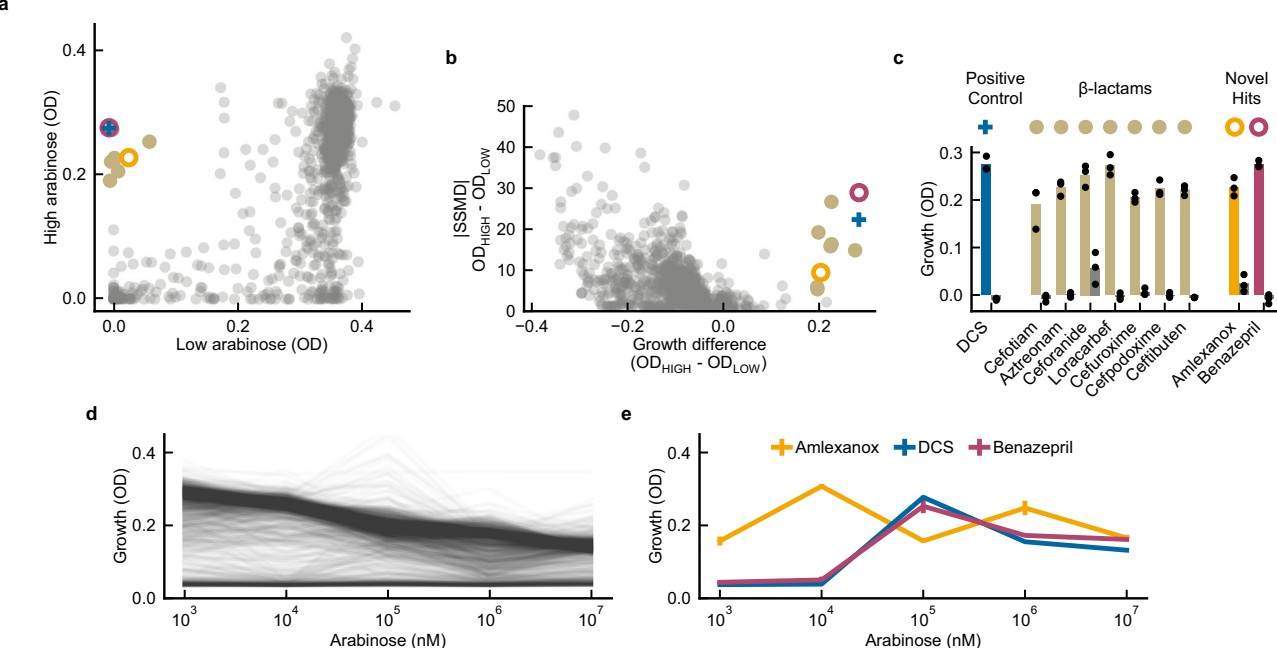

**Fig. 2 A screen for targeted *Mtb* Alr inhibitors identifies benazepril. a** The TESEC *Mtb* Alr expression strain was induced with arabinose at high ($10^7$ nM) or low ($10^2$ nM) levels and treated with the 1280-compound Prestwick library at 0.1 mM. Growth was measured by OD after 8 h and the median of three biological replicates is plotted for each drug. Hit compounds inhibit the growth of low-Alr strains but not high-Alr strains, occupying the upper-left quadrant. **b** SSMD scores were used to assess the statistical significance of the high- and low-induction growth levels. Hits were selected with an OD differential of 0.2 and an SSMD greater than 5. **c** Growth measurements for selected hits in high Alr (colored bars) and low Alr (gray bars) expression. Data are presented as the mean and individual data points for three biological replicates. **d** Chemical-genetic growth profiles of the TESEC *Mtb* Alr expression strain treated with each drug of the Prestwick library at 0.1 mM and a range of arabinose induction levels. **e** Selected chemical-genetic growth profiles for candidate hit compounds. Both DCS and benazepril showed growth inhibition only at low induction levels. Amlexanox, in contrast, did not show reproducible induction-specific activity. Data are presented as the mean and the standard deviation of three replicates. Source data are provided as a Source Data file.

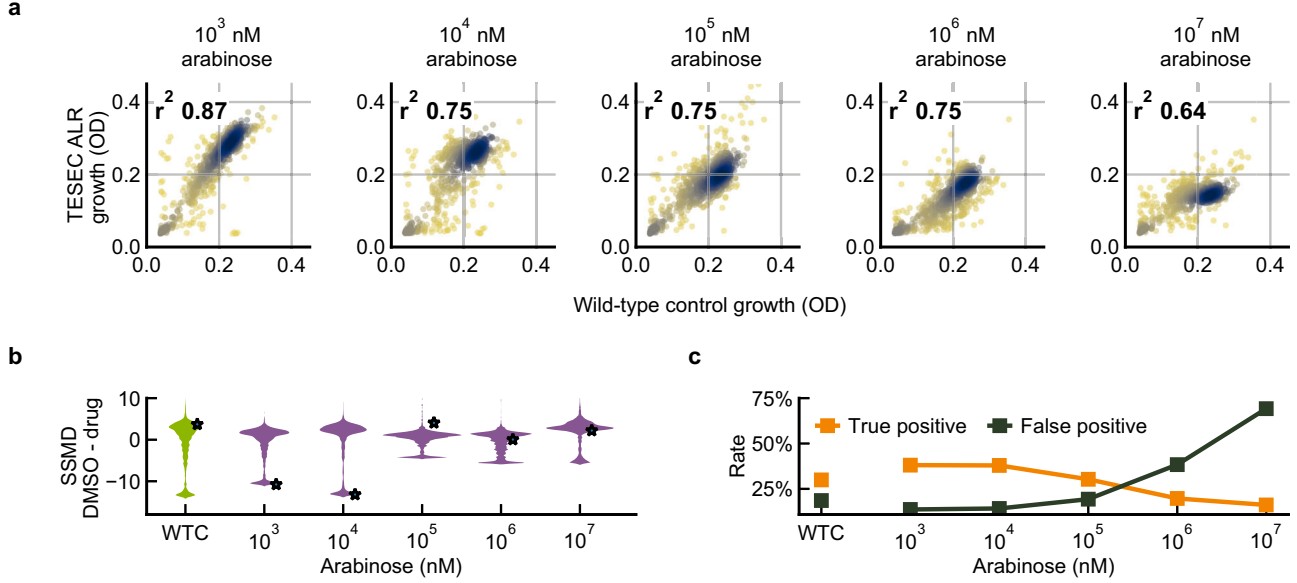

**Fig. 3 Chemical-genetic drug sensitivity is significantly altered by target overexpression. a** The growth effects of 1280 drugs from the Prestwick library on the TESEC *Mtb* Alr expression strain induced with varying levels of arabinose and the TESEC Alr+ wild-type control. Points are colored to indicate point density and $r^2$ is the Pearson correlation coefficient. **b** A violin plot of SSMD values comparing triplicate measurements of growth inhibition under drug treatment with growth measured in DMSO-only controls. High induction levels were associated with lower growth signal and higher noise resulting in less robust differentiation between inhibiting and non-inhibiting compounds. **c** Sensitivity of the TESEC assay in detecting the 183 known antibiotics in the Prestwick library. Predicted antibiotics were compounds showing more than 50% growth inhibition relative to DMSO controls. High induction levels were associated with lower growth, weaker signal, and more false positives. Source data are provided as a Source Data file.

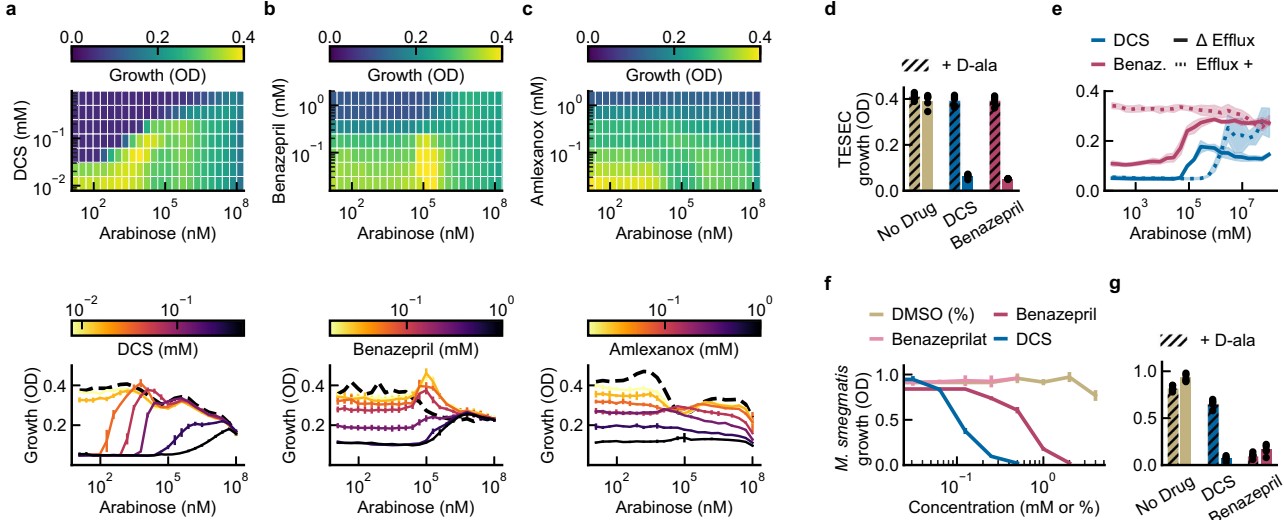

**Fig. 4 Characterization of hits with TESEC chemical-genetic profiling and in *M. smegmatis*. a–c** The TESEC *Mtb* Alr expression strain was induced at a range of arabinose concentration and exposed to the indicated drugs at a range of concentrations, producing a two-dimensional growth profile for each drug. Heatmaps of growth under DCS and benazepril treatment revealed a characteristic interaction between drug activity and target expression, appearing as a diagonal line. Amlexanox did not show this interaction. Detailed dose-response profiles were qualitatively different for each drug. Data are presented as the man and $CI_{95}$ for three biological replicates. **d** Supplementation of TESEC *Mtb* Alr with 5 mM D-alanine, the enzymatic product of Alr, eliminated the inhibitory activity of both 0.25 mM DCS and 1 mM benazepril. Data are presented as the mean and Individual data points for five biological replicates. **e** Restoring the activity of the TolC efflux system eliminated the inhibitory effect of 0.5 mM benazepril, but not of 0.1 mM DCS. Lines represent the mean and shaded areas the $CI_{95}$ of six biological replicates. **f** *M. smegmatis* growth was inhibited by benazepril and DCS, but not benazeprilat or DMSO-only controls, in the millimolar range. Data are presented as the mean and $CI_{95}$ of five biological replicates. **g** Supplementation with 5 mM D-alanine rescued growth of *M. smegmatis* treated with 0.25 mM DCS, but not cells treated with 1 mM benazepril. Data is presented as the mean and individual data points for five biological replicates. Source data are provided as a Source Data file.

from live cells (Fig. 4e). The importance of efflux activity was anticipated in our original design concept for TESEC screening, in which efflux pumps were inactivated to expand the range of discoverable drugs.

**Whole-cell activity of benazepril in *M. smegmatis* and *M. tuberculosis*.** We next sought to characterize the effect of benazepril as an anti-mycobacterial. Benazepril showed inhibitory activity against the non-pathogenic model mycobacterium *M. smegmatis* mc$^2$155, producing an $IC_{50}$ of 1 mM (Fig. 4f). The $IC_{50}$ for DCS was 0.1 mM under the same conditions. Spot plating assays confirmed this activity to be bactericidal as well as growth inhibitory (Supplementary Fig. 2).

Within the human body benazepril undergoes hydrolysis to produce the active ACE inhibitor benazeprilat[33]. This form did not show activity against either mycobacterial strain up to the limit of solubility, 0.5 mM. Supplementation with 5 mM D-alanine was able to rescue the growth of *M. smegmatis* in the presence of 0.25 mM DCS. The effect of 1 mM benazepril, in contrast, was not rescued (Fig. 4g). This may indicate that benazepril interacts with additional mycobacterial targets beyond Alr.

The H37Rv strain of *Mtb* was growth-inhibited by benazepril at concentrations between 2.25 and 4.5 mM, as compared to 125 μM for DCS (Table 1). As with *M. smegmatis*, D-alanine supplementation did not rescue this effect.

**Benazepril is a noncompetitive inhibitor of Alr.** The in vitro kinetics of benazepril inhibition were examined with a two-step biochemical reaction (Fig. 5). Purified Alr converts D-alanine to L-alanine which, in turn, is used by L-alanine dehydrogenase to reduce NAD+ to fluorescence-measurable NADH[34]. Control assays confirmed that benazepril does not inhibit the L-alanine

**Table 1 Antibiotic activity of benazepril against *Mtb*. *M. tuberculosis* H37Rv was cultured in 7H9 medium with the indicated treatment.**

| Compound | Dilution range | MIC |
|---|---|---|
| DCS | 0–2.0 mM | 0.125 mM |
| DCS + 5 mM D-ala | 0–2.0 mM | >2 mM |
| Benazepril | 0–4.5 mM | 2.25–4.5 mM |
| Benazepril + 5 mM D-ala | 0–4.5 mM | 2.25 mM |
| Benazeprilat | 0–0.5 mM | >0.5 mM |
| Isoniazid | 0–30 μM | 0.3 μM |
| DMSO | 0–5% | >5% |

Metabolic viability was measured with the resazurin reduction assay after 7–9 days. The activity of known *Mtb*-inhibitor DCS was rescued by the addition of D-alanine, consistent with targeted inhibition of Alr. The first-line *Mtb* drug isoniazid served as a positive control. Observed values were consistent across five biological replicates.

dehydrogenase reporter enzyme, making the assay suitable for characterizing the benazepril-Alr interaction (Supplementary Fig. 3).

Half-maximal inhibition of Alr activity was achieved with 4 μM DCS or 300 μM benazepril when D-alanine was supplied at 3 mM (Fig. 5a). Varying the concentration of D-alanine yielded conventional Michaelis–Menten kinetics for Alr (Fig. 5b) with benazepril affecting the maximum enzyme velocity, $V_{MAX}$, but not the D-alanine affinity constant, $K_M$ (Fig. 5c). These results are consistent with a noncompetitive inhibition mechanism for benazepril, unlike the competitive action of DCS[35,36].

**TESEC screening extends to other metabolic targets.** In developing TESEC, we sought to create a flexible chemical-genetic assay able to accommodate many potential targets. We tested the

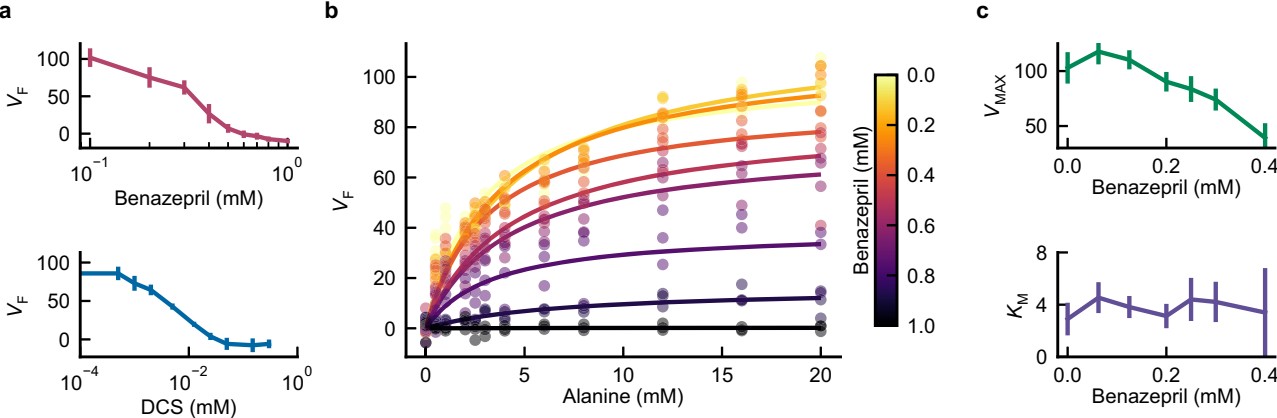

**Fig. 5 Benazepril inhibits *Mtb* Alr in vitro.** His-tagged *Mtb* Alr was purified and incubated with the indicated drug concentrations. Alr activity, $V_F$, was assayed using a coupled reaction yielding NADH. **a** Inhibition curves for benazepril and DCS. Data is presented as the man and $CI_{95}$ of three independent assays. **b** In vitro kinetics of Alr varying both inhibitor and substrate. At least three individual measurements are plotted for each pair of concentrations. Lines indicate best-fit Michaelis–Menten curves for each tested benazepril concentration. **c** Best-fit values of Michaelis–Menten parameters as a function of benazepril concentration. Data is presented as the mean and $CI_{95}$ for best-fit parameter values, as estimated using Student's t-distribution. Source data are provided as a Source Data file.

modularity of our system by constructing and screening additional TESEC strains (Fig. 6). We chose 4 enzymes required for amino acid biosynthesis and proposed as anti-mycobacterial drug targets: Asd[37], CysH[38], DapB[39], and TrpD[40].

For each target, we deleted the *E. coli* homolog and expressed the codon-optimized enzyme from *Mtb* in the standard TESEC vector. The resulting strains showed diverse dose-responses to target induction and unique induction optima (Fig. 6a–d). Following the strategy used for Alr, we selected high and low arabinose induction levels for differential screening.

We performed high-throughput screening of the Prestwick library in triplicate for all five targets (Fig. 6e–h) and selected hit compounds causing differential growth ($OD_{HIGH} - OD_{LOW} > 0.1$) at significant levels (SSMD > 5).

Hits were identified for Asd and DapB (Fig. 6i, h). All hit compounds were small and hydrophilic relative to the set of known antibacterials annotated in the Prestwick library (Fig. 6k). This may reflect the role of the *E. coli* membrane as a selective barrier to activity in the TESEC assay. No hits were identified in the CysH and TrpD strains, which showed poor growth at all induction levels and high variance during screening. These results highlight the importance of precise control and low-burden heterologous expression for robust chemical-genetic drug discovery.

As with benazepril, we validated the hit compounds by creating chemical-genetic growth profiles varying both drug concentration and arabinose induction. Four of the five hit compounds produced characteristically diagonal growth profiles, confirming a relationship between target abundance and drug sensitivity (Fig. 6l–p). This relationship was also evident in single arabinose dose-response curves for selected drug levels in the micromolar range: arabinose induction of the TESEC strain, but not a wild-type control, significantly increased resistance to the hit compound (Fig. 6q–u). A fifth hit, diethylstilbestrol, showed no interaction with target expression in follow-up assays.

Two of the four confirmed hits have previously shown activity against whole-cell mycobacteria in the micromolar range: pentamidine[41] and bromhexine[42]. Both drugs are administered through inhalation and show high bioavailability in the lungs[43,44]. Deoxycorticosterone has been shown to reduce mycobacterial growth within human macrophages[45]. For each of these compounds, our results suggest a previously unknown metabolic anti-microbial mechanism of action. Riluzole has no demonstrated

anti-mycobacterial activity to our knowledge, although it contains a trifluoromethoxy group rare in medicinal chemistry and found in many recent anti-TB clinical candidates including delamanid, pretomanid and, telacebec[46].

Of the five hit compounds, only pentamidine showed activity against *M. smegmatis* in the micromolar range (Supplementary Fig. 4). In contrast bromhexine, deoxycorticosterone and riluzole showed reproducible activity in the TESEC strain but not against *M. smegmatis*. This discrepancy may be due, in part, to membrane efflux activities present in wild-type cells but deleted in TESEC strains.

To better characterize the role of membrane efflux in TESEC screening, we tested our hit compounds against *E. coli* with wild-type efflux activity, as well as strains bearing deletions in the TolC system and five additional efflux deletion strains (Supplementary Fig. 5). For four compounds (Benazepril, bromhexine, diethylstilbestrol riluzole), the deletion of specifically *tolC* was necessary to reveal drug sensitivity. In contrast, varying the efflux phenotype had only modest effect on responses to the other two drugs, pentamidine and deoxycorticosterone, suggesting that their signal in our differential screen was produced mainly by the engineered variations in target abundance. These results are consistent with the original design motivations for the TESEC platform, in which efflux-deficient *E. coli* were engineered to reveal target-specific interactions rather than whole-cell activities.

## Discussion

This study demonstrates the use of TESEC screening to identify an anti-*Mtb*, anti-Alr activity of benazepril. The Prestwick drug library used here has been screened extensively for antibiotic activity, including in whole-cell assays against live *Mtb*[41], and targeted assays against purified Alr[47]. The extraction of a lead from this well-explored resource suggests that TESEC screening passes compounds through a selective filter unlike that of other discovery technologies.

The activity of benazepril against live mycobacteria appeared to differ mechanistically from that in the TESEC assay. In the *E. coli* model, the inhibitory effect of benazepril could be rescued by supplementation with D-ala, the enzymatic product of Alr, consistent with an activity specific to that enzyme. In *M. smegmatis* and *Mtb*, D-ala supplementation did not rescue benazepril inhibition, suggesting benazepril may act on other targets as has been

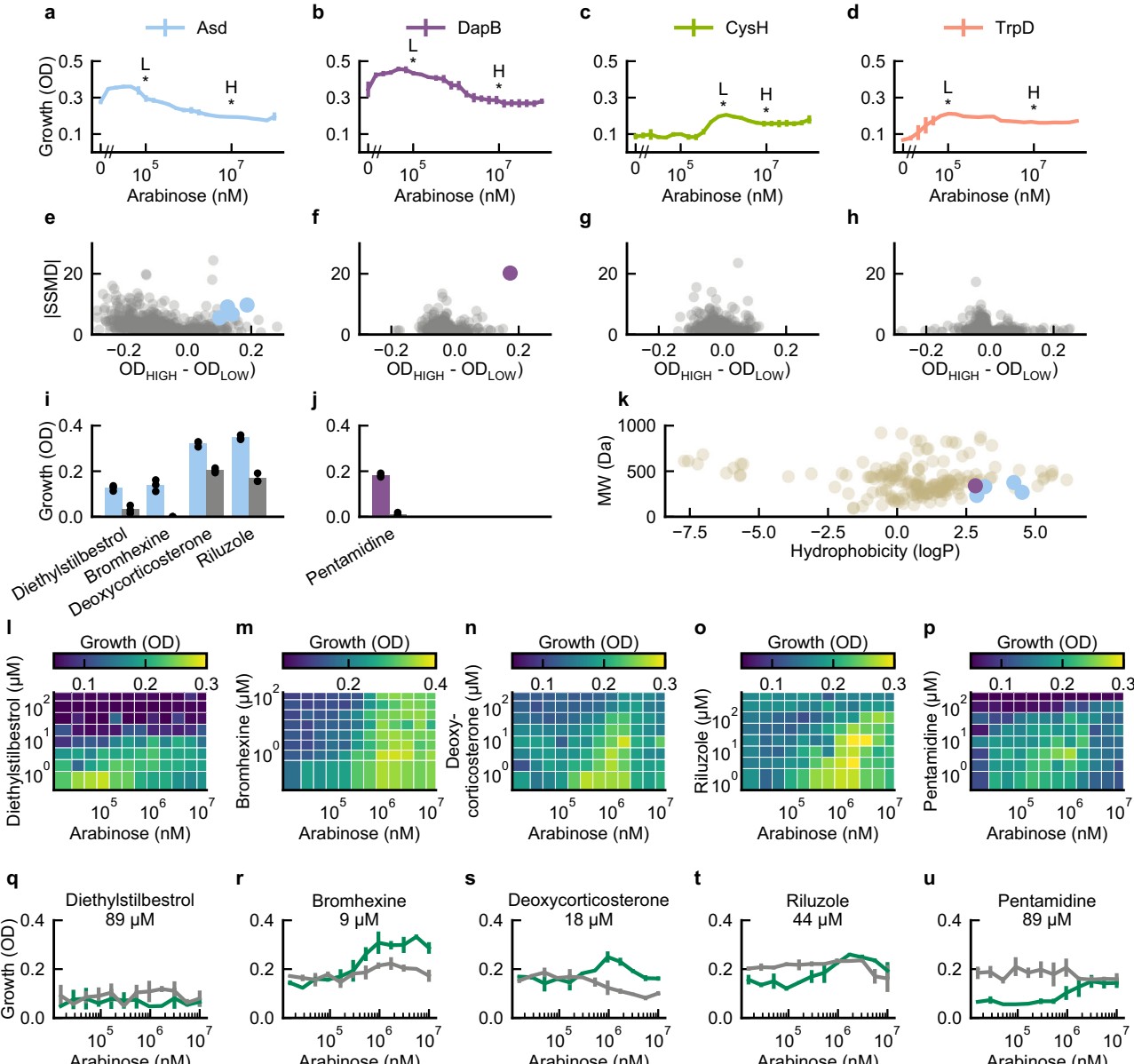

**Fig. 6 TESEC supports screening for diverse metabolic targets. a–d** TESEC strains for the indicated metabolic targets grown on defined medium with the essential enzyme induced with a range of arabinose concentrations. The indicated low (L) and high (H) arabinose concentrations were selected for differential screening. Data are presented as the mean and $CI_{95}$ of six biological replicates. **e–h** Differential screening of the Prestwick library was performed at both low and high arabinose concentrations in triplicate. Hits were selected as compounds with median differential OD measurements of 0.1 and SSMD scores >5. **i, j** Hit compounds identified for targets Asd, and DapB. No significant hits were identified for CysH or TrpD. Colored bars indicate mean growth under high induction, gray bars at low induction. Individual data points are shown for three biological replicates. **k** Molecular weight and hydrophobicity scores for identified hit compounds (Asd, blue; DapB, purple) and for all annotated antibacterials in the Prestwick library (beige). Hydrophobicity scores were estimated with the alvaMolecule software package. **l–p** Chemical-genetic validation profiles show TESEC grows as a function of both drug dose and TESEC target induction level. With the exception of diethylstilbestrol, hit compounds showed the characteristic diagonal pattern indicating that drug sensitivity depended on target expression level. **q–u** Individual dose–response curves extracted from the chemical-genetic profiles for a range of arabinose concentrations and the indicated drug dose. Except diethylstilbestrol, drug-treated TESEC strains showed improved growth at higher arabinose induction (green lines). Wild-type controls, in which target expression was not controlled by arabinose, did not show improved growth (gray lines). Data are presented as the mean and $CI_{95}$ for three biological replicates. Source data are provided as a Source Data file.

shown for cycloserine[48]. Benazepril is also likely to be effluxed, as indicated by its substantially higher MIC in mycobacteria (~2 mM) relative to the efflux-negative TESEC strain (~100 μM). This highlights a trade-off in TESEC similar to that in purified biochemical assays: bypassing the role of the membrane to reveal target-specific interactions in early-stage screening means the membrane may reappear as a point of failure at later stages.

The notoriously impermeable mycobacterial envelope represents a challenge for TESEC screening as it does for other antimycobacterial discoveries strategies. Target-based screens present no membrane challenge and often produce impermeant hits as a result[49]. Whole-cell screens against *Mtb* are plagued by redundant hits against promiscuous membrane targets that are inherently more accessible than cytosolic targets[50,51]. The TESEC assay

includes a bacterial membrane but not a mycobacterial membrane. How might this substitution impact screening results?

As with other Gram-negative bacteria, the *E. coli* outer membrane is coated with a stiff layer of lipopolysaccharides that excludes most large molecules[52]. Small molecules must be sufficiently hydrophilic to pass through outer membrane porins, then sufficiently hydrophobic to cross the inner lipid bilayer by solubility-diffusion[52]. This triple filtration means that most Gram-negative antibiotics occupy a narrow physico-chemical space[53].

Although mycobacteria are classified as Gram-positive, their envelope resembles that of Gram-negative strains in some respects, notably in the presence of an outer membrane bridged by porins[25]. The mycobacterial envelope is exceptionally rich in lipids and includes a layer of waxy long-chain mycolic acids[54]. These properties may explain why anti-*Mtb* actives tend to be smaller and more lipophilic than antibiotics in general[55].

Any compound identified by TESEC screening then confirmed for anti-*Mtb* activity will need to pass through both bacterial membranes. Benazepril's small size (MW 424) compares to that of many known antibiotics[56]. By hydrophobicity (clogP 1.3), benazepril is at the high end for Gram-negative penetration[56] and the low end for *Mtb*[55].

Once entering the cell a would-be antibiotic must contend with a battery of efflux pumps that are numerous and diverse in both *E. coli*[57] and *Mtb*[58]. We did not attempt to use *E. coli* as a model for efflux in *Mtb*. Instead, we eliminated major efflux activities by deleting *tolC*, making TESEC screening a mostly efflux-agnostic assay. This choice was motivated by the complexity and context-dependence of efflux in *Mtb*, which varies among natural isolates[59] and during the course of an infection[60]. For example, a study of 21 clinical isolates of *Mtb* found 10 strains with increased expression of at least one putative efflux pump relative to the H37Rv reference strain, with 6 different pumps showing natural variation[61]. In this context, even a whole-cell screen with live *Mtb* cannot fully capture the clinical diversity of the *Mtb* membrane barrier. The lack of a good laboratory model for *Mtb* efflux remains a bottleneck for antibiotic development at all stages[62].

Why perform a drug screen using an *E. coli* model rather than directly in the pathogen of interest? The TESEC approach offers low material costs, technical simplicity and base-level biosafety requirements. Synthetic biology tools developed for *E. coli* allow precise gene expression control, which our results indicate is important to minimize non-specific physiological stress. Efforts to make non-model microbes more genetically tractable may enable TESEC-like screening in other hosts[14]. Alternately, genetic alterations of *E. coli* may allow it to more closely emulate the physiology or membrane structure of other microbes[63], leading to the convergence of the two approaches.

Beyond the scientific challenges, low-cost technologies like TESEC may help to alleviate the economic pressures that limit antibiotic development[20]. Early-stage screening represents a significant fraction of total development costs due to high failure rates that require many screens to be launched for each advancing compound[64]. While traditional pharma companies are increasingly unwilling to assume all of the financial risks[65], emerging economic models propose to distribute the costs and risks among academic laboratories, small- and medium- sized enterprises, charitable organizations and national research institutes in high-burden countries[21,66–68]. TESEC strains can be freely replicated and exchanged, supporting highly multilateral projects characteristic of open science[69] and open drug discovery[70,71].

## Methods

### Construction of the TESEC host strain
The TESEC host was derived from BW25113[72]. Deletions of the genes *araC*, *tolC*, *entC*, *alr*, *dadX*, *dapB*, *asd*, *cysH*, and *trpD* were introduced by phage transduction of kanamycin resistance cassettes from the Keio collection[73], with cassettes subsequently removed by transient expression of Flp recombinase[72]. A complete list of strains used in this study is provided in Supplementary Table 1.

### Construction of the TESEC expression system
The plasmids pRD123 and pRD131[28] were modified for compatibility with Golden Gate assembly[18]. Type IIS restriction sites were removed and a lacZ cassette was introduced at the cloning site to facilitate blue/white screening.

Coding sequences for expressed proteins were ordered as synthetic DNA (Integrated DNA Technologies) and codon-optimized for expression in *E. coli*. The *Mtb* Alr sequence was obtained from the *Mtb* H37Rv genome as gene ID Rv3423c[23].

Four additional *Mtb* target sequences of interest *dapB* (Rv2773c), *asd* (Rv3708c), *cysH* (Rv2392), *trpD* (Rv2192c) were taken from the same *Mtb* genome.

A complete list of plasmids created for this study is provided in Supplementary Table 2. Plasmid DNA sequences are available as a Supplementary Data file.

### *E. coli* culture conditions
Conventional strain manipulations used standard Luria-Bertani (LB) media and 1.5% agar LB plates. Experiments were performed using defined medium using an M9 base: 11.26 g/L M9 Minimal Salts (BD 248510), 2 mM $MgSO_4$, 0.1 mM $CaCl_2$, and 4 g/L D-fructose as the primary carbon source. To support robust growth, the base media was supplemented with 1 g/L complete amino acids mix (USBiological D9516) and 30 mg/L vitamin B1.

Plasmids carrying the KanR and AmpR markers were maintained with 50 μg/mL kanamycin and 100 μg/mL ampicillin, respectively. Growth media for strains bearing the *alr* and *dadX* deletions were supplemented with 100 μg/mL D-alanine except when performing Alr-dependent growth assays. All incubations were performed at 37 °C with shaking. All OD measurements were collected at 600 nm.

### Mycobacterial culture conditions
*M. smegmatis* mc²155 was grown at 37 °C in Tryptic Soy Broth (TSB, Sigma 22092) supplemented with 0.5% Tween 80 (Sigma P5188).

*M. tuberculosis* H37Rv was grown at 37 °C in Middlebrook 7H9 broth (Sigma M0178) containing 0.2% glycerol and 10% OADC supplement (BD 211886).

### Fluorescence measurements of GFP-tagged *Mtb* Alr
The expression of GFP-tagged Alr was measured in defined medium supplemented with D-alanine to eliminate the growth requirement of Alr expression.

An arabinose dilution series was prepared beginning with 1 M arabinose and proceeded in 22 steps of two-fold dilution with a final well receiving no arabinose. A Tecan Freedom Evo liquid-handling robot was used to dispense 30 μL of each solution into a 384-well deep-well plate (Corning 3342).

The TESEC GFP-tagged *Mtb* Alr strain was grown overnight in defined medium without arabinose. The following morning cells were resuspended in defined medium at an OD of 0.05. 30 μL of cell suspension was added to each well of the arabinose dilution series. Cells were incubated for one hour. GFP fluorescence was measured from a minimum of 10,000 cells using the FITC-A channel of a flow cytometer (BD LSRFortessa).

### High-throughput screening
Drug screens were performed in two formats: 96-well and 384-well. The initial drug screen was performed in 96-well format using the TESEC *Mtb* Alr expression strain under high- and low-induction conditions. Subsequent screens were miniaturized to 384-well format and applied a wider range of induction conditions.

For 96-well format screens, the 1280 compound Prestwick Chemical Library was prepared with each compound at 10 mM concentration in DMSO. 1.5 μL samples were aliquoted to microplates with clear flat bottoms (Greiner 655090) using a Tecan Freedom Evo liquid-handling robot. The leftmost and rightmost columns of each plate were reserved for DMSO-only control wells.

Cells were grown overnight in defined medium supplemented with D-alanine. The next day cells were washed three times by centrifugation and resuspension in phosphate buffered saline (PBS), then diluted to an OD of 0.05 in defined medium lacking D-alanine but supplemented with arabinose at 0.1 μM for low Alr induction or 10 mM for high Alr induction.

Screening plates were filled with the dilute cell suspension at a volume of 150 μL, producing final drug concentrations of 0.1 mM and 1% DMSO. One OD reading was collected prior to incubation to establish the background absorbance of drugs alone. Plates were sealed with aluminum foil and incubated for 10 h. Aluminum foil was removed prior to taking post-incubation OD measurements with a microplate reader (Tecan Infinite).

384-well format screens were performed with small-volume microplates (Greiner 784101). An acoustic liquid handler (Labcyte Echo 500) was used to dispense 150 nL droplets of library drugs at 10 mM in DMSO. Cells were washed as above and diluted into defined media at an OD of 0.05. The diluted cell suspension was dispensed to the screening plates at a 15 μL volume to produce a final drug concentration of 0.1 mM and 1% DMSO. Plates were sealed with aluminum foil and incubated with shaking for 10 h. Foil was removed prior to final OD measurements.

Screening hits were identified by comparing mean differential OD values and SSMD scores calculated in Python. A single compound, Chicago Sky Blue 6B, was excluded from statistical analysis because of high background absorbance.

**Chemical-genetic profiling of hit compounds**. A two-dimensional dilution series of arabinose and the indicated drug were prepared in defined media using a Tecan Freedom EVO liquid-handling robot. Overnight cultures of the TESEC *Mtb* Alr expression strain were grown in defined media supplemented with D-alanine, then washed in PBS and diluted in D-alanine-free medium. 50 µL of washed cells were dispensed at a final OD of 0.05 into clear, flat-bottom, 384-well plates (Thermo-Fisher 242764) containing the two-dimensional dilution series. OD measurements were collected after 10 h of incubation.

The effect of TolC-mediated efflux was investigated with TolC+ and EntC+ control strains derived from the TESEC ancestor BW25113[74]. Overnight cultures were grown in defined medium, washed in PBS, and resuspended at an OD of 0.05. 50 µL of washed cells were dispensed at a final OD6 of 0.05 into clear, flat-bottom 384-well plates (ThermoFisher 242764). The indicated drug concentrations were added from DMSO stock solutions with final DMSO concentrations kept below 1%. OD measurements were collected after 10 h of incubation.

D-alanine rescue experiments were performed using the TESEC *Mtb* Alr expression strain cultured and washed as above then diluted to OD of 0.05 in a defined medium supplemented with 5 mM D-alanine. OD measurements were collected after 10 h of incubation.

**Growth inhibition assays with *M. smegmatis* mc²155**. A dilution series was prepared for each drug in TSB + 0.5% Tween 80. Drugs stocks for benazepril (Sigma B0935), benazeprilat (USP 1048641), and DCS (Sigma C6880) were prepared in DMSO, with final DMSO concentrations kept constant at 1% across all assays.

Single colonies of *M. smegmatis* mc²155 were grown to saturation for 48 h in TSB + 0.5% Tween 80. Drug treatments were prepared as 2 mL cultures in the same medium with cells diluted to OD 0.01, then incubated for 24 h with shaking. An additional 0.5% Tween 80 was used to solubilize cell aggregates prior to final OD readings.

**Growth inhibition assays with *M. tuberculosis* H37Rv**. Antibiotic activity against *Mtb* was determined using the microdilution method with five biological replicates. Two-fold serial dilutions of the tested drugs were prepared in a 96-well microplate. Live *Mtb* were inoculated with an initial density of $10^5$ Colony-Forming Units (CFUs) in growth medium at a volume of 200 µL.

The plates were incubated at 37 °C for 7–9 days, then 2.5 µg/mL resazurin (Sigma R7017) was added to each well before a further one-day incubation. Metabolic viability was determined visually by the color change of resazurin from blue to pink.

The effect of D-alanine on drug treatments was determined by further supplementing growth medium with a two-fold dilution series of D-alanine (0–5 mM).

**Expression and purification of *Mtb* Alr**. The *Mtb* Alr enzyme was expressed as the short variant characterized by Strych et al. that features a 24-residue N-terminal truncation relative to the annotated Rv3423c gene[23]. The enzyme was His-tagged and expressed in the TESEC Alr- Host strain, which has no other source of Alr activity.

The TESEC Alr Purification strain was grown to saturation in 500 mL of LB supplemented with ampicillin, kanamycin, D-alanine, and 100 mM L-arabinose. A cell pellet of approximately 2 grams was harvested by centrifugation, cooled to 4 °C and lysed with 10 mL B-PER reagent (ThermoFisher 78248) and a standard EDTA-free protease inhibitor cocktail (Merck 11873580001).

The lysate was cleared by centrifugation and passed over 3 mL HisPur spin columns (ThermoFisher 88226). Elution fractions were collected with increasing concentrations of imidazole (Sigma I2399) and checked for the presence of a single band using SDS-PAGE gels (Bio-Rad 4561081) and Coomassie staining. Successful elutions were combined and dialyzed (ThermoFisher 66380) against 20 mM Tris, 100 mM NaCl pH 8.0 for 72 h at 4 °C. Finally, samples were concentrated by centrifugal filtration in Amicon Ultra columns (Merck UFC9010). Total protein concentrations were determined using the Rapid Gold BCA Protein Assay (ThermoFisher A53226) calibrated to a standard curve following the manufacturer's protocol.

**_Mtb_ Alr inhibition assays**. The enzymatic activity of Alr was measured using a standard assay[34] in which L-alanine produced by Alr drives the stoichiometric production fluorescence-readable NADH through the activity of the coupling enzyme L-alanine dehydrogenase (Merck A7189).

Dilution series of benazepril (TCI Chemicals B3611) or DCS (Sigma C6880) were prepared in reaction buffer (20 mM Tris, 100 mM NaCl, pH 8.0) with 10% DMSO. 5 µL solutions of 90 nM purified *Mtb* Alr in reaction buffer were incubated with 5 µL drug treatments for 30 min at room temperature. After pre-incubation, 35 µL reaction cocktails of 0.05 U/mL L-alanine dehydrogenase and 20 mM β-NAD (Sigma N8129) in sodium bicarbonate buffer (pH 10.5) were added to each well.

The reactions were started with the addition of 5 µL of D-alanine solution (0–20 mM) and followed by the evolution of fluorescence (340 nm excitation, 460 nm emission) using a TECAN Spark plate reader. Final DMSO concentrations were kept constant at 1% for all reactions.

Initial reaction rates were determined using the linear portion of the fluorescence evolution curves. Alr activity is reported as a percentage relative to the initial velocity in 20 mM D-alanine and no inhibitor. Best-fit kinetic parameters were determined by nonlinear regression in Python.

**Culture conditions for additional TESEC strains**. TESEC strains for *trpD* and *cysH* were cultured in the same defined M9-base medium used for TESEC *Mtb* Alr. To convey growth dependence on the *Mtb* enzymes, the complete amino acid supplement was replaced with tryptophan dropout powder (USBiological D9530) for *trpD* or cysteine and methionine dropout powder (USBiological D9537-08) for *cysH*.

TESEC strains for *asd* and *dapB* were cultured in M9-base medium supplemented with 1 g/L D-fructose, 1 g/L cysteine and methionine dropout powder, and 20 µg/mL diaminopimelic acid (DAP, Sigma D1377). Screening was performed in LB medium supplemented with 1 g/L D-fructose and 1 g/L complete amino acid supplement (USBiological D9516). DAP was omitted during growth curves and drug screening to convey growth dependence on enzyme expression.

**Reporting summary**. Further information on research design is available in the Nature Research Reporting Summary linked to this article.

## Data availability

Source data are provided with this paper. The data generated in this study have also been deposited in the Zenodo database (https://doi.org/10.5281/zenodo.6597306).

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

## Acknowledgements

We thank Jean-Emmanuel Hugonnet for advice on assays using live *Mtb*. Thanks to the Bettencourt Schueller Foundation long term partnership, this work was partly supported by the CRI Research Fellowship to EHW. Additional support was provided by MSDA-VENIR (DS-2018-0073).

## Author contributions

N.B., A.B.L., and E.H.W. conceived and planned the experiments. Z.E. performed the *Mtb* inhibition assays. A.A.A. and S.S.-C. contributed to strain development and high throughput screening for Alr. X.S. contributed to the data analysis. S.G., S.H., and J.S. developed and screened additional strains. N.B. performed the in vitro Alr biochemical assays. N.B., A.B.L., and E.H.W. developed the final manuscript.

## Competing interests

The authors declare no competing interests.
