## [Peer Review File · Nature Communications]

Reviewers' Comments:

Reviewer #1:

Remarks to the Author:

In this work, the authors present TESEC, a strategy of replacing an essential gene in E.coli with its heterogeneity homologous gene in Mtb, to create a low-cost platform for drug screening. Previously, similar approaches are made in previous research on either bacteria or mammalian cell drug screening, but the author first used gene-modified E.coli to screen anti-Mtb drugs. The authors constructed the TESEC cell and screened it with the Prestwick Chemical Library, a pool of 1280 chemicals. The author also found a potential drug molecule benazepril after screening. They also demonstrate how the strategy can be generalized to other targets.

The study is overall interesting and represents a unique application of synthetic biology. Conceptually, it's somewhat analogous to other strategies to simplify the experimental system to facilitate speed/efficiency of drug screening (e.g. the use of organoids for drug screening). My sense is the presented data are overall solid. The authors have also done thorough analysis in tuning the screening reliability (e.g. by examining the impact of the expression level of the target).

However, practically speaking, the central question is the extent by which this strategy would save time (in comparison to direct screening). As the authors demonstrated, the synthetic system needs substantial optimization in terms of the expression of the target and drug doses tested. I wonder if the fact that the screening only identifies one effective drug is a reflection that the platform happens to operate under a condition that favors the particular drug.

In the case of benazepril, it seems that the way it inhibits Mtb is mechanistically different from its inhibition on the TESEC system. This raises a major question regarding the effectiveness of the approach – I believe the authors should provide additional experimental evidence to show that. For a specific drug, this is not an issue. For the platform, however, there should be a reasonable expectation that the screening indeed significantly captures the drug-target interactions for the intended pathogen. Therefore, it's important to show that the screening can identify multiple drugs that can be evaluated in the 2nd stage (on Mtb). Testing the success rate of the leads is critical to demonstrate the efficacy of the approach (as well as its fundamental premise). It is odd that the authors did not follow on some other leads shown in Figure 6.

Other specific comments:

1. The authors made an interesting argument that direct screening will narrow the outcome to drugs that target membranes and miss cytosolic targets. The synthetic system may avoid this caveat. On the flip side, however, this argument raises a potential limitation of the screening. Though you can identify drugs that target cytosolic targets; these drugs may not necessarily be effective against Mtb. This critical question cannot be answered given the presented data, as only one drug was tested further with Mtb.

2. Figure 4e indicates that the effect of benazepril will be restored by efflux pump in E.coli. And in line 225 'This choice was motivated by the complexity and context dependence of efflux in Mtb strains, which varies among natural isolates and during the course of an infection', the author also admit that the efflux pump is complex and active in Mtb. Based on these, it is likely that benazepril will have different working concentration or sensitivity to different Mtb strains. At least more discussion should be added here about why Mtb H37Rv is chosen here (say the efflux pump type and strength), and data for the sensitivity of benazepril in multiple Mtb strains (better with different efflux pump type and strength) is strongly recommended.

Reviewer #2:

Remarks to the Author:

This paper by Bongaerts et al., describes a synthetic biology approach for identifying M. tuberculosis (Mtb) inhibitors by assaying known targets in E. coli. The idea of developing strains (called TESEC) where an essential E. coli enzyme is deleted and replaced with an equivalent target of heterologous origin can be useful for early drug screening efforts vs Mtb. The authors decide to

target alanine racemase (ALR), an enzyme involved in cell wall biosynthesis.

As a proof-of-concept study, the authors perform a screen of 1280 approved drugs and provide data revealing a single viable ALR inhibitor (benazepril; a known ACE inhibitor used against hypertension), which is then confirmed in vitro against both *M. smegmatis* and *Mtb*. Further biochemical assays indicated that that benazepril operated via a non-conventional non-competitive mechanism of inhibition (e.g., different than the competitive action of DCS). Four additional strains vs different *Mtb* metabolic targets involved in amino acid biosynthesis (i.e., *Asd*, *CysH*, *DapB*, and *TrpD*) were also tested and hits were identified for only *Asd* and *DapB*.

Overall, the strain design, characterization, and screen appear to be well performed and the data provided are robust. Thus, the work describes, convincingly, a new synthetic biology platform for identifying potential inhibitors of *Mtb* in vitro. However, in my view, this work at its current stage is too preliminary for a broad journal such as *Nature Communications* (and perhaps more suitable for a specialized journal), particularly due to the lack of cytotoxicity data and safety and efficacy studies in animals.

Specific comments:

- Was benazeprilat tested at concentrations similar to benazepril (up to 4.5 mM; Table 1)?
- Data presented (Fig. 4) indicates that benazepril may have additional targets beyond ALR. This is concerning due to the possibility of off-target effects.
- No toxicity data is shown for any of the hits.
- No safety and efficacy studies are conducted in animals for the hits identified.
- Other concerns include the low hit rate obtained from screening efforts and high doses needed for in vitro efficacy vs *Mtb* for the hits obtained (Table 1).

REVIEWER COMMENTS

Reviewer #1 (Remarks to the Author):

In this work, the authors present TESEC, a strategy of replacing an essential gene in E.coli with its heterogeneity homologous gene in Mtb, to create a low-cost platform for drug screening. Previously, similar approaches are made in previous research on either bacteria or mammalian cell drug screening, but the author first used gene-modified E.coli to screen anti-Mtb drugs. The authors constructed the TESEC cell and screened it with the Prestwick Chemical Library, a pool of 1280 chemicals. The author also found a potential drug molecule benazepril after screening. They also demonstrate how the strategy can be generalized to other targets.

The study is overall interesting and represents a unique application of synthetic biology. Conceptually, it's somewhat analogous to other strategies to simplify the experimental system to facilitate speed/efficiency of drug screening (e.g. the use of organoids for drug screening). My sense is the presented data are overall solid. The authors have also done thorough analysis in tuning the screening reliability (e.g. by examining the impact of the expression level of the target).

Thanks for this positive and fair summary.

However, practically speaking, the central question is the extent by which this strategy would save time (in comparison to direct screening). As the authors demonstrated, the synthetic system needs substantial optimization in terms of the expression of the target and drug doses tested. I wonder if the fact that the screening only identifies one effective drug is a reflection that the platform happens to operate under a condition that favors the particular drug.

In essence, the alr TESEC system identified two hits: D-cycloserine (a previously known drug) and benazepril. Given that the library used as proof-of-concept was heavily screened in the past, we find this result to be reassuring and would not have expected more hits. This is now confirmed by studying deeper results from the other targets where few or no hits are found. We argue that beyond the ease of implementing TESEC screening, it is thus highly

robust to false positive hits, highlighting the efficacy of the TESEC approach.

In the case of benazepril, it seems that the way it inhibits Mtb is mechanistically different from its inhibition on the TESEC system. This raises a major question regarding the effectiveness of the approach – I believe the authors should provide additional experimental evidence to show that. For a specific drug, this is not an issue. For the platform, however, there should be a reasonable expectation that the screening indeed significantly captures the drug- target interactions for the intended pathogen.

We have added text to the discussion section to make this important caveat clear to the reader.

The activity of benazepril against live mycobacteria appeared to differ mechanistically from that in the TESEC assay. In the *E. coli* model, the inhibitory effect of benazepril could be rescued by supplementation with D-ala, the enzymatic product of ALR, consistent with an activity specific to that enzyme. In *M. smegmatis* and *Mtb*, D-ala supplementation did not rescue benazepril inhibition, suggesting benazepril may act on other targets as has been shown for cycloserine⁴⁸. Benazepril is also likely to be effluxed, as indicated by its substantially higher MIC in mycobacteria (~2 mM) relative to the efflux-negative TESEC strain (~100 µM). This highlights a trade-off in TESEC similar to that in purified biochemical assays: bypassing the role of the membrane to reveal target-specific interactions in early stage screening means the membrane may reappear as a point of failure at later stages. (Page 14, Line 225)

Below, we describe follow-up assays for additional hit compounds that provide a more complete picture for how TESEC screening translates to activity in mycobacteria.

Therefore, it's important to show that the screening can identify multiple drugs that can be evaluated in the 2nd stage (on Mtb). Testing the success rate of the leads is critical to demonstrate the efficacy of the approach (as well as its fundamental

premise). It is odd that the authors did not follow on some other leads shown in Figure 6.

Following this recommendation, we have further characterized the leads shown in figure 6. To summarize the new data:

- Figure 6 now includes chemical-genetic profiles for each of the 5 new hit compounds. We measured TESEC strain growth while co-varying the drug dose and the target expression level, similar to our validation work on benazepril in figure 3. The resulting 2D heatmaps confirm a drug-target interaction for 4 of the 5 new hits.
- Supplementary figure S4 shows dose-response curves of the new hits on the growth of *M. smegmatis*. One of the 5 new hits (pentamidine) inhibits *M. smegmatis* growth in the micromolar range.

Including the hit compounds from the original screen, benazepril, cycloserine and amlexanox, we found in total 6/8 hits that validated in repeat TESEC assays and 3/8 that showed some whole-cell activity in mycobacteria. Although the sample size is small, we think these examples add value to the manuscript by allowing potential adopters of our system to develop some expectation for how it might perform in future screens against future targets.

The 6/8 validation rate in repeat TESEC assays is a solid performance in the often-messy world of high throughput screening. This is consistent with our other metrics showing good reproducibility in the triplicates of our screens (SSMD, Z-scores).

The 3/8 validation rate in whole-cell mycobacteria could be considered disappointing. But it is consistent with our assay design, which intentionally does not filter for whole-cell activity. By removing efflux pumps and hyper-sensitizing specific targets, our goal was to reveal novel target-specific leads. That some of these leads are effluxed, or not active at wild-type target abundance, is an expected consequence.

Revisions to the text on these points can be found in:

- Results section: "TESEC screening extends to other metabolic targets" (Page 11)
- Figure 6: "TESEC supports screening for diverse metabolic targets" (Page 12)
- Supplementary text section: "Effect of efflux activity on *E. coli* sensitivity to hit compounds" (Page S3)

- Supplementary figure S4: "Activity of the extended hit compounds in *M. smegmatis*" (Page S4)

Other specific comments:

1. The authors made an interesting argument that direct screening will narrow the outcome to drugs that target membranes and miss cytosolic targets. The synthetic system may avoid this caveat. On the flip side, however, this argument raises a potential limitation of the screening. Though you can identify drugs that target cytosolic targets; these drugs may not necessarily be effective against Mtb. This critical question cannot be answered given the presented data, as only one drug was tested further with Mtb.

Motivated by this comment, we expanded our investigation of the role of membrane efflux activity on the effectiveness of our hit compounds. The new data is presented in supplementary figure S5 and discussed in the results section "TESEC screening extends to other metabolic targets."

Unfortunately, we were not able to do these new experiments in Mtb. We used *M. smegmatis* as a first approximation of the Mtb membrane and efflux activities. Those results are discussed in response to the previous comment. Here, we used deletion mutants in *E. coli* to explore the effect of efflux activities on drug sensitivity.

There were three compounds that showed validated activity in our screen but did not kill whole-cell mycobacteria (Bromhexine, deoxycorticosterone, riluzole). Although every drug screening technique produces candidates that fail to advance, the most useful techniques also produce useful information about the mode of failure to guide future work. In the TESEC system, we can take advantage of *E. coli* as a tractable host to develop additional chemical-genetic assays to characterize our lead compounds.

For figure S5, we extended the TESEC system to report on specific efflux activities associated with resistance to our hit compounds. We individually deleted 5 different efflux systems, in addition to the *toIC* deletion used for our main screening strain. For each efflux mutant, we measured growth as a function of the concentration of benazepril and five additional hit compounds. For 5 of 6 drugs, the *toIC* deletion was essential to reveal drug sensitivity. This

is consistent with that gene's role in multiple broad-specificity efflux systems. In the case of pentamidine, the deletion of *mdtK* also marginally increased sensitivity. The other genes tested did not appear to have a role effluxing these compounds.

Interestingly, deoxycorticosterone was not active against *E. coli* of the parent strain even with the deletion of *tolC*. Only the full TESEC strain, in which *tolC* was deleted and the target enzyme Asd was expressed at low levels, was sensitive to deoxycorticosterone. This suggests that resistance to that drug in wild-type *E. coli* is mediated not by efflux but by the relatively high expression of the target in the wild-type background.

These results suggest a future direction for chemical-genetics in the TESEC system: testing drug sensitivity in a range of host backgrounds to explore mechanisms of resistance. Like the TESEC system, characterizing drug interactions in this way could be done cheaply and safely. We added the following text to put these results in context.

Of the five hit compounds, only pentamidine showed activity against *M. smegmatis* in the micromolar range (Sup Fig S4). In contrast bromhexine, deoxycorticosterone and riluzole showed reproducible activity in the TESEC strain but not against *M. smegmatis*. This discrepancy may be due, in part, to membrane efflux activities present in wild-type cells but deleted in TESEC strains.

To better characterize the role of membrane efflux in TESEC screening, we tested our hit compounds against *E. coli* with wild-type efflux activity, as well as strains bearing deletions in the TolC system and five additional efflux deletion strains (Sup Fig S5). For four compounds (Benazepril, bromhexine, diethylstilbestrol riluzole), the deletion of specifically *tolC* was necessary to reveal drug sensitivity. In contrast, varying the efflux phenotype had only modest effect on responses to the other two drugs, pentamidine and deoxycorticosterone, suggesting that their signal in our differential screen was produced mainly by the engineered variations in target abundance. These results are consistent with the original design motivations for the TESEC platform, in which efflux-deficient *E. coli* were engineered to reveal target-specific interactions rather than whole-cell activities. (Page 14, Line 205)

2. Figure 4e indicates that the effect of benazepril will be restored by efflux pump in *E. coli*. And in line 225 'This choice was motivated by the complexity and context

dependence of efflux in *Mtb* strains, which varies among natural isolates and during the course of an infection', the author also admit that the efflux pump is complex and active in *Mtb*. Based on these, it is likely that benazepril will have different working concentration or sensitivity to different *Mtb* strains. At least more discussion should be added here about why *Mtb* H37Rv is chosen here (say the efflux pump type and strength), and data for the sensitivity of benazepril in multiple *Mtb* strains (better with different efflux pump type and strength) is strongly recommended.

We expanded the discussion section to describe the challenge of the *Mtb* membrane. Even whole-cell screens against the H37Rv reference strain of *Mtb* are an imperfect proxy for the diverse efflux activities in clinical *Mtb* strains. TESEC screening is a worse approximation than classical whole-cell *Mtb*, but probably a better one than target-based screens using purified enzymes.

In our view, the future of early stage drug discovery probably doesn't look like a single assay that perfectly represents clinical conditions. Instead, multiple assays will be required, each one telling a partial story of compound's target, permeability, stability etc. In this context, the strength of TESEC screening is in simplicity and versatility. Although we can't represent the *Mtb* membrane directly, we can systematically vary efflux activities in TESEC strains. This would lead to a more complete picture of a compound's tendency to efflux and through which mechanisms, and then to a general strategy for evading efflux across many *Mtb* clinical isolates.

Once entering the cell a would-be antibiotic must contend with a battery of efflux pumps that are numerous and diverse in both *E. coli*⁵⁵ and *Mtb*⁵⁶. We did not attempt to use *E. coli* as a model for efflux in *Mtb*. Instead we eliminated major efflux activities by deleting *tolC*, making TESEC screening a mostly efflux-agnostic assay. This choice was motivated by the complexity and context-dependence of efflux in *Mtb*, which varies among natural isolates⁵⁷ and during the course of an infection⁵⁸. For example, a study of 21 clinical isolates of *Mtb* found 10 strains with increased expression of at least one putative efflux pump relative to the H37Rv reference strain, with 6 different pumps showing natural variation⁵⁹. In this context, even a whole-cell screen with live *Mtb* cannot fully capture the clinical diversity of the *Mtb* membrane barrier. The lack of a good

laboratory model for *Mtb* efflux remains a bottleneck for antibiotic development at all stages⁶⁰.

Why perform a drug screen using an *E. coli* model rather than directly in the pathogen of interest? The TESEC approach offers low material costs, technical simplicity and base-level biosafety requirements. Synthetic biology tools developed for *E. coli* allow precise gene expression control, which our results indicate is important to minimize non-specific physiological stress. Efforts to make non-model microbes more genetically tractable may enable TESEC-like screening in other hosts¹⁴. Alternately, genetic alterations of *E. coli* may allow it to more closely emulate the physiology or membrane structure of other microbes⁶¹, leading to the convergence of the two approaches. (Page 16, Line 245)

Reviewer #2 (Remarks to the Author):

This paper by Bongaerts et al., describes a synthetic biology approach for identifying *M. tuberculosis* (Mtb) inhibitors by assaying known targets in *E. coli*. The idea of developing strains (called TESEC) where an essential *E. coli* enzyme is deleted and replaced with an equivalent target of heterologous origin can be useful for early drug screening efforts vs Mtb. The authors decide to target alanine racemase (ALR), an enzyme involved in cell wall biosynthesis.

As a proof-of-concept study, the authors perform a screen of 1280 approved drugs and provide data revealing a single viable ALR inhibitor (benazepril; a known ACE inhibitor used against hypertension), which is then confirmed in vitro against both *M. smegmatis* and Mtb. Further biochemical assays indicated that that benazepril operated via a non- conventional non-competitive mechanism of inhibition (e.g., different than the competitive action of DCS). Four additional strains vs different Mtb metabolic targets involved in amino acid biosynthesis (i.e., Asd, CysH, DapB, and TrpD) were also tested and hits were identified for only Asd and DapB.

Thanks for this positive and fair summary.

Overall, the strain design, characterization, and screen appear to be well performed and the data provided are robust. Thus, the work describes, convincingly, a new synthetic biology platform for identifying potential inhibitors of Mtb in vitro. However, in my view, this work at its current stage is too preliminary for a broad journal such as Nature Communications (and perhaps more suitable for a specialized journal), particularly due to the lack of cytotoxicity data and safety and efficacy studies in animals.

Specific comments:

- Was benazeprilat tested at concentrations similar to benazepril (up to 4.5 mM; Table 1)?

Benazeprilat was tested up to the limit of solubility. We have clarified this in the text:

Within the human body benazepril undergoes hydrolysis to produce the active ACE inhibitor benazeprilat. This form did not show activity against either mycobacterial strain up to the limit of solubility, 0.5 mM. (Page 10, Line 153)

- Data presented (Fig. 4) indicates that benazepril may have additional targets beyond ALR. This is concerning due to the possibility of off-target effects.
- No toxicity data is shown for any of the hits.
- No safety and efficacy studies are conducted in animals for the hits identified.
- Other concerns include the low hit rate obtained from screening efforts and high doses needed for in vitro efficacy vs Mtb for the hits obtained (Table 1).

For these screens we used the Prestwick chemical library, a collection of approved and off-patent drugs suitable for repurposing. All the drugs discussed in this work are therefore safe for use in humans. Nevertheless, many have other indications that would make them unsuitable for most TB patients. Benazepril, for example, is used to treat hypertension.

We think that the use of this library also explains the low hit rate from our screen. Prestwick is a well-explored resource that includes many known antibiotics (which kill both high- and low- expression TESEC strains so are not identified as differential hits) and many drugs previously tested for repurposing as antibiotics. This perspective is highlighted in the text:

This study demonstrates the use of TESEC screening to identify a novel anti-*Mtb*, anti-ALR activity of benazepril. The Prestwick drug library used here has been screened extensively for antibiotic activity, including in whole-cell assays against live *Mtb*(32), and targeted assays against purified ALR(33). The extraction of a new lead from this well-explored resource suggests that TESEC screening passes compounds through a selective filter unlike that of other discovery technologies. (Page 15, Line 290)

Regarding the suitability of this work for a general audience, we would ask the reviewers not to judge the impact primarily by how far we can advance a clinical candidate. Instead, we offer TESEC as a novel and generalizable chemical-genetic assay format. Differing substantially from existing target-based and whole-cell screens, TESEC adds a new dimension to the way drugs are characterized. TESEC's versatility, low cost, low biosafety requirements, and the ease with which TESEC strains can be shared and re-used, make it a useful tool for synthetic biologists and anyone working in early-stage drug discovery.

Reviewers' Comments:

Reviewer #1:

Remarks to the Author:

The authors have satisfactorily addressed my comments.

Reviewer #2:

Remarks to the Author:

The authors have addressed my prior comments. Thus, I recommend acceptance of the paper at this stage.

In this round of revision, the reviewers have not asked us to address specific points.

REVIEWER COMMENTS

Reviewer #1 (Remarks to the Author):

The authors have satisfactorily addressed my comments.

Reviewer #2 (Remarks to the Author):

The authors have addressed my prior comments. Thus, I recommend acceptance of the paper at this stage.